# A Comparative Study of Fitness Levels among Norwegian Youth in 1988 and 2001

**DOI:** 10.3390/sports7020050

**Published:** 2019-02-22

**Authors:** Kari Aaberge, Asgeir Mamen

**Affiliations:** 1Faculty of Education, Arts and Sports, Western Norway University of Applied Sciences, 6856 Sogndal, Norway; kari.aaberge@hvl.no; 2Institute of Health Sciences, Kristiania University College, 0105 Oslo, Norway

**Keywords:** adolescents, physical fitness, oxygen uptake, jump height, flexibility

## Abstract

We compared the fitness levels of cohorts of 15-year-old youth in 1988 and 2001 to ascertain whether there was a negative trend in fitness. The subjects were 15-year-old boys and girls from the same geographical area, n = 192 in 1988 and n = 191 in 2001. They participated voluntarily and could leave the project whenever they wished. The following variables were used to assess fitness: Maximal oxygen uptake, jump height, shoulder flexibility, and hamstring flexibility. Maximal oxygen uptake was estimated with submaximal ergometer cycling, jump height by the Sargent jump-and-reach test, shoulder flexibility as the distance between thumbs when doing straight-arm backwards circling while holding a broomstick, and hamstring flexibility by an active straight-leg-raise test. Differences between groups and quartiles were analyzed by Gosset’s (Student’s) t-test, using a significance level of 0.05. The two cohorts did show different levels of physical fitness. The 1988 group was 3.9 cm better on jump height and 4.2 cm better on shoulder flexibility, while the 2001 group had 3.3° better hamstring flexibility. The lowest performing quartile did less well in 2001 on oxygen uptake and jump height. We recommend an increased focus on improving fitness in low-performing adolescents.

## 1. Introduction

Several studies have reported a secular decline in physical fitness among youth. Tomkinson et al. found a global decline of 0.43% per year in running performance from 1980 to 2000 when analyzing data from 11 countries [1]. In 2007, Tomkinson and Olds found a performance decline of 0.36% per year in running performance with data from 27 countries [2]. Westerstahl et al. [3] detected a 3% decline in aerobic fitness among Swedish adolescents from 1974 to 1995. Eisenmann and Malina [4] found that directly measured peak oxygen uptake had been relatively stable from 1938 to 2000 from age 6 to 18. The exception was girls from 15 to 18 years old, who showed a decline of 20% over the last decade. Fredriksen et al. [5] also found no changes in average fitness values among 8- to 16-year-old Norwegian children in 1998 compared with average fitness values of children from Norway and Sweden from 1952 onwards. However, the variance was larger in 1998, indicating a polarization of the results. In another study from Norway, Dyrstad, Berg, and Tjelta found a decrease in 3000 m running performance from 1969 to 2009 for 15-to 18-year-old adolescents [6]. When comparing male conscripts’ test results from 1980–1985 with test results from a 2002 cohort of conscripts, Dyrstad, Aandestad and Hallén (2005) found a reduced mean maximal oxygen uptake in the 2002 group. This reduced mean result was due to an increased number of low performance results and a reduced number of high performance results among the men from 2002 [7]. A decrease in cardiorespiratory fitness between 1992 and 1999 and between 1992 and 2012 was observed by Dos Santos et al. [8] for both sexes. Westerstahl et al. [3], using the Sargent jump-and-reach test, found that girls had a poorer score in 1995 than in 1974. Boys, on the other hand, had a better score in 1995 than in 1974. Ellingsen [9] found that all scores on physical variables he measured among boys (aged 15) were lower in 1997 than in 1968. Girls, however, scored better on some of the strength exercises, movement, and oxygen uptake in 1997 than in 1968. Huotari et al. [10] looked at secular trends in muscle fitness in young Finns from 1976 to 2001 and found a small increase in muscular fitness in 2001 compared with 1976 for both boys and girls, but the differences were small, with effect sizes in the range of 0.17–0.20. However, the differences between the best and the worst had increased in 2001. Loprinzi [11] found no change in handgrip strength over four years, 2011–2014, in a population of Americans. Huotari et al. [12] measured what they called ‘fundamental movement skills’ in 15-year-olds in 2003 and 2010. The sum of the indexes of the three skills did not change, but for both sexes, the coordination score declined in 2010, whereas for girls, an improvement in control skill was seen from 2003 to 2010. Dos Santos et al. [8] found Mozambican children in 1992 to be more flexible, faster, and more agile than in 2012; here, boys’ handgrip strength increased from 1992 to 2012, whereas that of the girls decreased. Bianco et al. argue for an Enriched Sport Activities program because of the reported decline in physical activity among adolescents [13]. The picture is thus not at all clear, and there are good reasons to investigate the trend of ‘fitness’, more broadly defined, among adolescents. For that reason, the aim of this study was to compare the fitness levels among 15-year-old boys and girls from 1988 with those of a similar age group from 2001, from the same geographical region, with the same test battery and the same test leader (see the Section 2), to determine whether ‘fitness’, defined as ‘endurance, muscle strength, and flexibility’, had changed.

## 2. Materials and Methods

The data in this study are from two cross-sectional collections from 1988 and 2001 on 15-year-old boys and girls from Sogn og Fjordane County in Norway. The same test leader, Kari Aaberge, conducted both studies and used much of the same equipment. In 1988 the tests were conducted in April, while in 2001 the tests took place in January to March. This minor difference in time is considered by us to be insignificant, as all tests were performed indoors. A small difference in age between the two groups is also thought to be without significance.

### 2.1. Participants

All 15-year-old boys and girls in Sogn og Fjordane County were invited to take part in the study. In 1988, from a total of 281 persons, 230 persons volunteered (82% of the total), for whom we have a data set of 192: 106 boys and 86 girls. In 2001, 254 persons volunteered (79% of the total, N = 320). Here we have a data set of 191: 99 boys and 92 girls.

### 2.2. Equipment

In 1988, the subjects cycled on a Tunturi W1 stationary bicycle (Tunturi OY, Turku, Finland) or on a Bodyguard 990 (Jonas Øgland A/S, Sandnes, Norway). In 2001, the same Bodyguard 990 ergometer and one Monark 824E stationary bicycle were used (Monark AB, Varberg, Sweden). We correctly calibrated both bicycles before testing. The heart rate monitors were from Polar (Polar OY, Kempele, Finland): In 1988 the Sport Tester (PE 4000) and in 2001 the Accurex+. Body mass was measured with the same Seca column scale (Seca GmbH, Germany) both times. In 2001, the scale was calibrated against another Seca scale (Seca 770) and found to be valid and reliable. For shoulder flexibility, a broomstick and a standard measuring tape were used to measure the distance between hands. A custom-made protractor was used to measure the angle between the horizontal surface and the lifted, extended leg at both test times in the straight-leg-raise test.

### 2.3. Protocol

#### Indirect and Direct Tests of Maximal Oxygen Uptake

We used the submaximal Åstrand-Ryhming test [14] for estimation of maximal oxygen uptake. The test started when the participants were able to maintain a cadence of 50 RPM. The participants worked for a minimum of 6 min. If, after 6 min, the heart rate differed more than five bpm from the value at the 5th min, the test was prolonged one or two minutes more to reach steady state. Steady-state heart rate, the mean of the last two minutes of the test, was used to estimate the maximal oxygen uptake. This result was then corrected with a correction factor based on the participant’s age [15].

Fifteen (eight boys and seven girls) of the 192 participants in 2001 also had their oxygen uptake measured directly. We used a MetaMax II metabolism analyzer (Cortex Biophysik, Leipzig, Germany). The participants cycled on the Monark bicycle at 75 RPM. The load was increased by 30 W (0.4 kg) every minute until voluntary exhaustion. Maximal values were the median of the three consecutive highest 10 s values at the end of the test.

### 2.4. Sargent Jump-and-Reach Test for Jump Height

We used the Sargent jump-and-reach test [16,17]. The participants placed themselves close to the wall and marked baseline height by lifting their arm maximally and putting a mark on the wall. Then they jumped as high as possible and set a new mark on the wall at maximal height. The jump was from a standing position with a quick knee bend and arm swing before taking off. The difference between the start point and top point was recorded as the jump height. This test has been found by de Salles et al. to be reliable among 13- to 16-year-old [18].

### 2.5. Broomstick Test for Shoulder Flexibility

The participants held a broomstick in front of them in a comfortable grip with arms straight. Then they raised the broomstick over their heads and lowered it behind their heads, keeping their arms straight while adjusting (sliding) the handgrip to allow for shoulder rotation. Once the broomstick was positioned above their hips, the distance between their thumbs was measured and taken as the shoulder flexibility. The results were not corrected for shoulder width.

### 2.6. Straight-Leg-Raise Test for Hamstring Flexibility

The participants started the test lying with their backs on the floor and legs straight out. Then they lifted one straight leg up as far as possible, keeping the knee joint straight and with ankle dorsiflexion. The vertex of the custom-made protractor was placed as close as possible to the back of the leg and the angle between the leg and a horizontal line to the floor was measured and used as the hamstring flexibility [19].

### 2.7. Ethical Considerations

In 1987, a description of the project was sent to the Primary and Secondary School Council and the municipal school councils in Sogn og Fjordane County. After approval from all parts, the principal and the PE teachers of each school was contacted and informed about the project. The principal then took responsibility to inform the pupils and parents about it. The testing was conducted during school time, as part of the PE lessons, but participation was voluntary. Except for the application to the Primary and Secondary School council, which had been closed down in 1992, the procedure was the same in 2000. The study design and data collection does not require approval from Regional Ethical Committee in Norway or Norwegian Centre for Research Data as no person identifiable information were recorded. In Norway the Health Research Act was put in force in 2009, long after our data collection. In 1987, the approval of research was within the institution, later the Regional Ethical Committees were established (REC). REC does not comment on previous data collections. The Norwegian Data Protection Service states that as long as no person identifiable data are collected, there is no requirement for notification (http://www.nsd.uib.no/personvernombud /en/notify/notification_test.html).

### 2.8. Statistics

Descriptive results are presented as means, standard deviations (SDs), and percentiles, unless otherwise noted. Group means are compared with Gosset’s (Student’s) t-test. Quantiles were compared using quantile regression, with quantiles 0.1 to 0.9 with a 0.1 increment. A significance level of 0.05 was applied. Statistical software used was SigmaPlot 14 (Systat Software GmbH, Erkrath, Germany), Winks 7.0.9 (TexaSoft, Cedar Hill, TX, USA), SPSS 24 (IBM SPSS, Armonk, NY, USA) and Stata IC 15.1 (Stata Corp. LLC, College Station, TX, USA).

## 3. Results

### 3.1. Maximal Oxygen Uptake

The estimated mean maximal oxygen uptake showed normal values for boys and girls, being close to 50 mL·kg^−1^·min^−1^ for the boys and more than 40 mL·kg^−1^·min^−1^ for the girls. See Figure 1 and Table 1. In 2001, the SD was larger for both boys and girls than in 1988. The 25th percentile was identical for boys in 1988 and 2001 and was 3 mL·kg^−1^·min^−1^ lower for girls in 2001. The 75th percentile was 2.5 mL·kg^−1^·min^−1^ higher in 2001 for boys, whereas girls differed by only 1 mL·kg^−1^·min^−1^, with girls from 2001 highest. The directly measured maximal oxygen uptake for the 15 participants (eight boys and seven girls) was 46.8 ± 11.1 mL·kg^−1^·min^−1^. This did not significantly differ from the estimated values, 43.4 ± 9.5 mL·kg^−1^·min^−1^ (*p* = 0.06). Moreover, the correlation between the two measures was 0.56.

### 3.2. Jump Height

From Table 2, we can see that the average jump height for boys dropped almost 4 cm from 1988 to 2001. The girls demonstrated an even larger reduction in jump height, close to 7 cm in the same period. The lower performance of the girls was evident throughout the whole performance range, see Figure 2. The 2001 boys, on the other hand, did not differ greatly from the 1988 boys at the 100th percentile. It is illustrative of the development that the 25th percentile in 1988 (boys) had a value close to the boys’ mean in 2001.

### 3.3. Shoulder Flexibility

In Figure 3 and Table 3, we can see that the results indicate a significant stiffening of the shoulder muscles from 1988 to 2001 for both sexes. In 2001, the boys had to keep their hands 4 cm further apart compared with the 1988 mean results. The girls’ performance is even more indicative of stiffer muscles since the mean result increased nearly 6 cm from 1988 to 2001.

### 3.4. Hamstring Flexibility (Straight-Leg Raise)

The mean hamstring flexibility was better in 2001 than in 1988 for both boys and girls as seen from Figure 4. The 25th percentile was 5° better in 2001 for boys and girls, for the right leg. For the left leg, girls improved 2.5° and boys 5°. At the 10th percentile (Figure 4), the 1988 and 2001 groups were almost equal for boys and girls. At the 90th percentile, the 2001 group showed better results than the 1988 group did. Especially the girls improved: The 2001 group had a 17° increase in hamstring flexibility compared with the flexibility of the 1988 group. For both boys and girls, the 80th percentile showed the largest difference between 1988 and 2001. See Table 4 for details.

### 3.5. Quantile Analysis

In Table 5, the significant results of the quantile regression are presented.

We analyzed the worst and the best 1/4 of the groups, as Q_worst_ and Q_best_. The mean and the SD of these groups were compared across gender and year. For the lowest-performing quantile (Q_worst_), maximal oxygen uptake, jump height, and shoulder flexibility decreased significantly for both boys and girls. The 2001 group excelled in the left straight-leg-raise test for boys and girls. The quantile of best performance (Q_best_) had the 2001 boys best on oxygen uptake and on right and left straight-leg-raise tests. Shoulder flexibility was not statistically different between the 1988 and 2001 groups. For girls, oxygen uptake did not differ between year groups. The 1988 group was better on jump height and shoulder flexibility. Hamstring flexibility was superior in the 2001 group. See Table 6 for details.

## 4. Discussion

We found differences in fitness levels between the two youth groups from 1988 and 2001. Analysis of the means showed that the 1988 group surpassed the 2001 group in jump height and shoulder flexibility for boys. Only jump height was statistically significantly different between the two girl groups. The boys and girls in 2001 had better performance on the straight-leg-raise test. Also, boys performed better than girls on endurance capacity and jump height, while girls were better on the straight-leg-raise test.

### 4.1. Oxygen Uptake: Endurance

Much focus has been placed on the development of endurance among children, and several studies by Tomkinson [2,20,21] have documented a downward secular trend in endurance on a global scale. This trend has diminished in recent years [20]. In Sweden, the study by Westerstahl et al. [3] documented a decrease in running performance from 1974 to 1995. In 1974, boys had a median result on a run-walk test of 2131 m. In 1995, the result was 2027 m, about 5% lower. The decrease for girls was about 2%. Ellingsen [9] noted an improvement in girls’ running performance from 1968 to 1997, and a deterioration for boys. Such an increase for girls is not surprising, as health and sport performance for girls have been strongly promoted since 1968. Eisenmann and Malina [4] collected results of directly measured oxygen uptake in the 20th century, and found the values to be relatively stable for boys and young girls. In 1930–1940, boys 13–16 years old showed a peak oxygen uptake of 2.22 L·min^−1^ or 45.6 mL·kg^−1^·min^−1^. In the 1990s the result was 2.74 L·min^−1^/53.4 mL·kg^−1^·min^−1^. Peak values were found in the 1970s, at 3.41 L·min^−1^/54.6 mL·kg^−1^·min^−1^. Girls peaked in the same decade at 2.39 L·min^−1^/41.6 mL·kg^−1^·min^−1^ and were in the 1990s back to values from the 1960s. This is somewhat surprising, as female sport has developed greatly since the 1970s, both in number of participants and in performance level. The mean value we found for boys was lower than that found by Eisenmann and Malina [4]. The 15-year-old girls in our sample had a higher oxygen uptake than the levels from the 1980s and 1990s in Eisenmann and Malina’s [4] study.

Differences in measuring methods might explain some of this divergence, as argued by Keren et al. [22], and the question of representativeness is always important to consider. Our mean results did not differ significantly between 1988 and 2001, in agreement with Fredriksen et al. [5], but their 1998 results are superior to ours. Boys had a peak oxygen uptake, directly measured by maximal treadmill running, of 61 (5) mL·kg^−1^·min^−1^, while girls had 49 (6) mL·kg^−1^·min^−1^. These values are significantly higher than the boys’ 1988 and 2008 values. Our 1988 results for girls are significantly lower than the results for Fredriksen et al.’s girls [5]. As in their study, we found an increasing polarization across time, as the gap between the best and the worst increased. This may be more worrisome than just a decrease of the mean, because the low-performing subjects today are now in even poorer health and may in the near future need help from the health system. According to Kodama et al. [23], for each MET (3.5 mL·kg^−1^·min^−1^) that oxygen uptake diminishes, all-cause mortality increases 13% in adults. To have a small elite that improves in fitness is not enough to compensate for the increased challenge represented by those who are least fit, as Dyrstad, Aandestad and Hallén have shown [7].

The difference between best and worst performance among boys in 1988 was 20 mL·kg^−1^·min^−1^. In 2001 this gap had increased to 31 mL·kg^−1^·min^−1^. The same trend can be seen for girls. In the Fredriksen et al. study [5], the SD was about 5 mL·kg^−1^·min^−1^ for the groups tested. The studies they used for comparison showed smaller SDs, about half the size. This increased variation can be caused by a number of factors: Less physical activity in 1998 than in 1952 (the oldest data collection), and increased body mass in 1998. Fredriksen et al. [5] also mentioned the possibility of selection bias to explain their results. Andersen et al. [24] comment on the difficulty of comparing results from different investigations. Differences in test leader experience were found to influence the results, and test protocol and ergometers may also affect the results. The 15 students that also performed a direct maximal test had a correlation coefficient of 0.56 to the estimated results, close to that found by Kasch [25]. He concluded that the indirect test was inadequate as a substitute for a direct measurement. A much better agreement between direct and indirect results was found by Teräslinna et al., 0.92 [26]. Even with results like these, it is important to carefully evaluate individual results from any indirect test of aerobic fitness.

### 4.2. Jump Height

The values shown here are lower than the normative values presented by Mackenzie [27]. For boys aged 16–19, their ‘average’ value is 40–49 cm; this value corresponds roughly to that of the 90th percentile in both the 1988 group and the 2001 group in our study. The girls performed even worse in 2001; the average value 90th percentile is lower than the normative average value of Mackenzie [27]. Taylor et al., Bovet et al., and Temfemo et al. have also found better values for the same age group [28,29,30].

Tomkinson [31] found that power ability had increased in adolescents from 1960 until 1990, and then declined, but was still higher in 2003 than in 1958. It can be difficult to compare jump height results, because the tests used might influence the results. We used the common Sargent test, used since 1921. Tests done on force platforms might give other results, because the jump technique is not quite the same. A vertical jump requires considerable anaerobic power, and is thus sensitive to body mass, and there are reports of an increase in body mass, especially in fat mass, in modern youth [3,32,33]. The reason for the poorer performance in 2001 might therefore be higher body mass compared to the 1988 cohort, but the relevant 1988 data are now lost. In the study by Westerstahl et al. [3], they found poorer results for girls and better results for boys from 1974 to 1995. The girls decreased non-significantly by about 3%, but the boys’ jumping height increased significantly, by ~7%. This difference was still significant after adjusting for body dimension, as the 1995 cohort was heavier and had a greater BMI.

### 4.3. Shoulder Flexibility

Frequent keyboard use can cause muscle damage in the shoulders because of static load. This type of repetitive strain injury is also known as ‘work-related upper limb disorder’. Rotator cuff syndrome and frozen shoulder are the most frequent injuries of this type. It might be plausible that extensive computer use and mobile phone texting can predispose for such diagnoses. They might also reduce shoulder range of movement (ROM) even if other symptoms are missing. Although we have not recorded computer use time in the groups, computers were probably used by many of the 2001 group, but none of the 1988 group. If we compensate for shoulder width, by subtracting 35 cm for girls and 38 cm for boys, the results show that all groups had poor flexibility compared with reference data from Mackenzie [27]. The average distance between thumbs for boys in 1988 was 21 cm greater than the reference value [27], and for girls, 14 cm greater. In 2001, the results are even poorer; the difference is 25 cm for boys and 19 cm for girls. We cannot find a simple explanation for this poor result. It might be caused by selection bias or randomness or by differences in how the test was performed.

### 4.4. Straight-Leg-Raise Test

This test had better values for boys and girls in 2001. Good flexibility in the hamstring has been found to decrease lower extremity overuse injuries in adults [34]. We can therefore call this a generally positive trend. According to Shacklock [35], normal ROM values for adults are from 45° to 80°. The results we have found are thus in the upper range. For athletes, too, values similar to ours have been found by López-Miñarro [36]. Reduced muscular strength might give increased ROM in a joint. If the 2001 group was less physically active than the 1988 group, reduced muscle strength could be one explanation for the result. The poor results from the jump-and-reach test also point to reduced muscle strength as an explanation. However, the rather similar passive straight-leg-raise test (Lasegue’s test) has low specificity, thus limiting its diagnostic accuracy, according to a systematic review by Devillé et al. [37]. This might also be true for the active test we used and thus could explain some of the differences.

### 4.5. Strengths and Weaknesses

The strengths of this study are the high percentage that took part, that the test leader was the same both times and that time-proven tests were used in the fitness assessment. The weaknesses are that indirect estimation may have a bias, manual measurements may be inaccurate and motivation among the cohorts may have differed.

## 5. Conclusions

The average values for maximal oxygen uptake did not differ greatly, but the difference between the best and the poorest performances increased in both sexes. Average jump height performance decreased significantly for both boys and girls from 1988 to 2001. Reduced leg muscle strength might be a reason for this development, as hamstring flexibility in the same period increased significantly in both 2001 groups. Shoulder flexibility was better in 1988 for boys, with no significant difference for the girls. Increased computer time may be a reason for the findings. The increased polarization seen in all variables is a strong indicator that a greater focus is needed on the health of low-fitness youth in particular. Schools should have a particular responsibility for this, as the school system reaches all adolescents.

## Figures and Tables

**Figure 1 sports-07-00050-f001:**
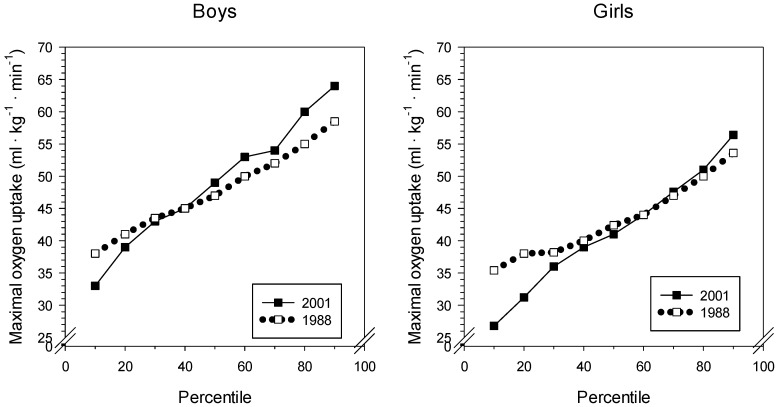
Percentiles of maximal oxygen uptake for boys and girls from 1988 and 2001.

**Figure 2 sports-07-00050-f002:**
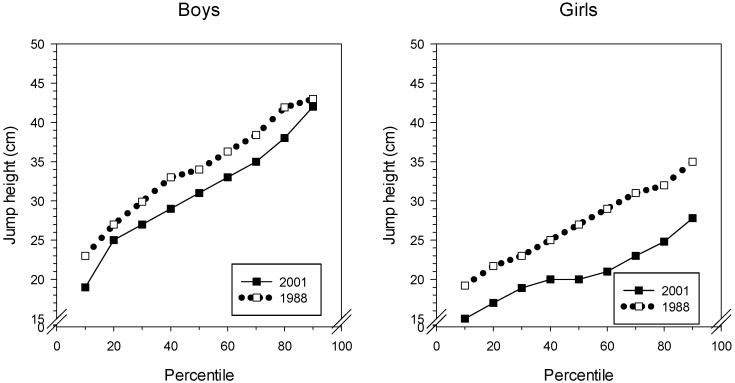
Percentiles of Sargent jump-and-reach test results for boys and girls from 1988 and 2001.

**Figure 3 sports-07-00050-f003:**
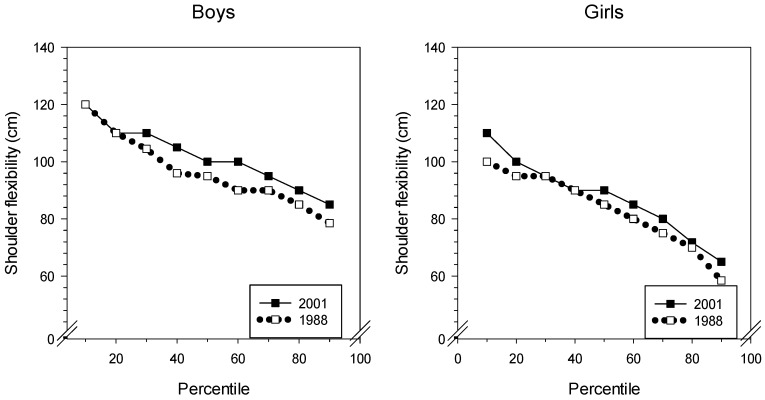
Percentiles of shoulder flexibility for boys and girls from 1988 and 2001.

**Figure 4 sports-07-00050-f004:**
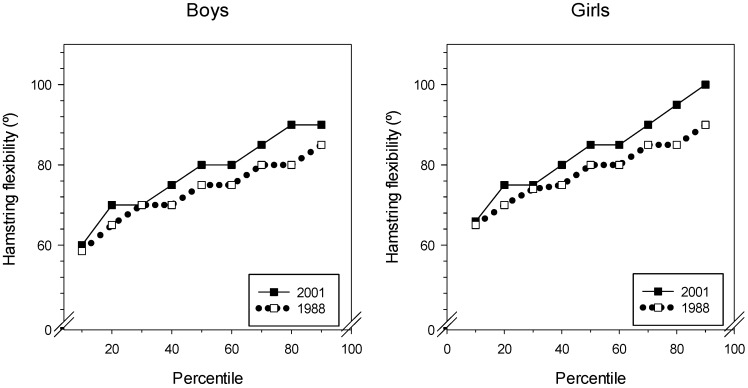
Percentiles of straight-leg-raise test, right leg, for boys and girls from 1988 and 2001.

**Table 1 sports-07-00050-t001:** Maximal oxygen uptake, higher values are better (mL·kg^−1^·min^−1^).

Cohort	n	Mean	SD	*p*	25th Percentile	75th Percentile
Boys 2001	99	49.0	12.0	0.332	42.0	56.0
Boys 1988	104	47.7	8.3		42.0	53.5
Girls 2001	77	42.0	11.3	0.374	35.0	49.0
Girls 1988	83	43.4	8.2		38.0	48.0

**Table 2 sports-07-00050-t002:** Jump height, higher values are better (cm).

Cohort	n	Mean	SD	*p*	25th Percentile	75th Percentile
Boys 2001	99	31.1	8.4	0.001	26.0	36.0
Boys 1988	106	35.0	7.9		30.0	41.5
Girls 2001	91	20.1	5.5	<0.001	16.0	24.0
Girls 1988	81	27.1	6.1		23.0	31.0

**Table 3 sports-07-00050-t003:** Shoulder flexibility, lower values are better (cm).

Cohort	n	Mean	SD	*p*	25th Percentile	75th Percentile
Boys 2001	99	100.0	14.6	0.039	110.0	90.0
Boys 1988	106	95.8	14.4		105.0	85.0
Girls 2001	92	87.2	17.2	<0.01	100.0	80.0
Girls 1988	86	81.6	16.0		95.0	75.0

**Table 4 sports-07-00050-t004:** Straight-leg-raise test, higher values are better (°).

Cohort		n	Mean	SD	*p*	25th Percentile	75th Percentile
Right Leg	Boys 2001	99	77.5	11.6	<0.001	70.0	80.0
	Boys 1988	106	72.2	9.0		65.0	85.0
	Girls 2001	92	83.1	13.1	0.004	75.0	90.0
	Girls 1988	85	77.9	10.6		70.0	85.0
Left Leg	Boys 2001	99	78.3	11.9	<0.001	70.0	85.0
	Boys 1988	98	71.6	8.4		65.0	75.0
	Girls 2001	92	82.5	13.8	0.036	72.5	90.0
	Girls 1988	85	78.6	10.5		70.0	85.0

**Table 5 sports-07-00050-t005:** Statistical significant results of quantile regression.

Variable	*τ*	*p*	Quantile
Boys shoulder flex	−2.12	0.036	0.1
Boys straight-leg-raise test, right leg	−2.99	0.004	0.9
Boys straight-leg-raise test, left leg	−7.85	<0.001	0.1
Boys straight-leg-raise test, left leg	−3.57	0.001	0.3
Boys straight-leg-raise test, left leg	−3.07	0.003	0.4
Girls maximal oxygen uptake	2.48	0.015	0.1
Girls maximal oxygen uptake	3.02	0.003	0.3
Girls maximal oxygen uptake	3.16	0.002	0.5
Girls maximal oxygen uptake	3.62	0.001	0.6
Girls jump height	−2.23	0.029	0.8
Girls straight-leg-raise test, right leg	2.00	0.048	0.7
Girls straight-leg-raise test, left leg	1.99	0.050	0.9

**Table 6 sports-07-00050-t006:** Comparison of the 1988 and 2001 groups, poorest performance (Q_worst_) and best performance (Q_best_). Results are mean for the worst 1/4 and best 1/4 of the sample, SD in brackets. On shoulder flexibility, lower values are associated with better performance. There was no correction for shoulder width in this test.

Quantile	Variable	Boys 2001	Boys 1988	Girls 2001	Girls 1988
**Q_worst_**	VO_2max_ (mL·kg^−1^·min^−1^)	34 (6)	36 (4) *	28 (4)	33 (3) *
	Jump height (cm)	20.3 (5.1)	24.6 (4.2) *	18.6 (5.6)	23.9 (6.1) *
	Shoulder flexibility (cm)	115.6 (5.8)	78.4 (7.6) *	88.2 (16.9)	71.3 (19.7) *
	Right straight-leg raise (°)	59.5 (4.1)	56.7 (3.8) *	64.8 (6.0)	62.0 (4.6) n.s.
	Left straight-leg raise (°)	61.9 (4.0)	58.1 (3.3) #	65.9 (5.6)	62.7 (3.3) #
**Q_best_**	VO_2max_ (mL·kg^−1^·min^−1^)	63 (7)	58 (4) #	57 (6)	54 (7) n.s.
	Jump height (cm)	41.1 (4.3)	44.7 (3.6) *	27.2 (3.2)	34.1 (4.0) *
	Shoulder flexibility (cm)	76.8 (10.3)	73.5 (6.5) n.s.	63.2 (18.4)	59.3 (11.7) *
	Right straight-leg raise (°)	91.0 (6.4)	82.0 (2.5) #	97.0 (9.5)	90.0 (5.9) #
	Left straight-leg raise (°)	90.7 (6.5)	81.9 (3.2) #	97.3 (9.1)	90.4 (5.9) #

* = 1988 group best (*p* < 0.05), # = 2001 group best (*p* < 0.05), n.s. = not significant.

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
