# Peer review of "A Comparative Study of Fitness Levels among Norwegian Youth in 1988 and 2001"

_sports, 2019, doi:10.3390/sports7020050_

Round 1
Reviewer 1 Report
Comments to the Author
This study compared the fitness level of a cohort of 15-year-old youth in 1988 with that of a cohort of youth of the same age in 2001 to see if there was a negative trend in the development of fitness level. Changes in maximal oxygen uptake, jump height, shoulder flexibility, and hamstring flexibility at the population level during this time period were calculated. Importantly, lowest performing quartile did less well in 2001 on oxygen uptake and jump height, for boys and girls. I have the following specific comments for the authors to consider.
Abstract, results: Can the same units for the declines be reported after matching on fitness.
Introduction: Throughout the introduction effects sizes of change in fitness are described per some papers. This is fine, but it is not clear how these results compare to those in the abstract. If percentile of change is the unit of choice in this field then what are the percentile of change in maximal oxygen uptake, jump height, shoulder flexibility, and hamstring flexibility observed in this study. Please give this unit of change in the abstract (and in the results if not given).
Methods: I like the changes in variability approach. However, it is not clear if the percentile curve is the most suitable approach. Why not use quantile regression to statistically test for percentile specific differences?
Results: Descriptions of the study samples are missing in the opening paragraph of the results (i.e., weight, height, BMI, etc.)
Discussion: Matching for fatness does not provide direct causal connection between changes in fatness and fitness. Please remove the any reference to causality.
Author Response
We thank You for the pertinent comments and have tried to implement them in the revised text:
I have the following specific comments for the authors to consider.
Abstract, results: Can the same units for the declines be reported after matching on fitness.
We have now included the actual difference between cohorts for the different exercises.
Introduction: Throughout the introduction effects sizes of change in fitness are described per some papers. This is fine, but it is not clear how these results compare to those in the abstract. If percentile of change is the unit of choice in this field then what are the percentile of change in maximal oxygen uptake, jump height, shoulder flexibility, and hamstring flexibility observed in this study. Please give this unit of change in the abstract (and in the results if not given).
In the introduction we report a variety of results from studies all over the world. In the Result chapter, and now in the Abstract, results are given in the same units.
Methods: I like the changes in variability approach. However, it is not clear if the percentile curve is the most suitable approach. Why not use quantile regression to statistically test for percentile specific differences?
Quantile regression was totally unknown to me. I have bought a one-year license for Stata IC 15.1 and performed the analysis, but are uncertain if this is the correct way or interpretation of the results, advice is most welcome! I have used Simultaneous quantile regression with bootstrap (20) and quantiles 0.1, 0.2, 0.3, 0.4, 0.5, 0.6, 0.7, 0.8, 0.9.
Results: Descriptions of the study samples are missing in the opening paragraph of the results (i.e., weight, height, BMI, etc.)
Quite right, that information is missing, and this is due to the fact that such data from the 1988 cohort is missing.
Discussion: Matching for fatness does not provide direct causal connection between changes in fatness and fitness. Please remove the any reference to causality
This is also true, but our intention is to point to a possible reason for the decline in jump performance, not anything else. Suggestion to re-write: ...and is thus sensitive to body mass, and there are reports of an increase in body mass, especially in fat mass, in modern youth [3,31,32]. The reason for the poorer performance in 2001 might therefore be higher body mass compared to the 1988 cohort, but the 1988 data are now lost.

Reviewer 2 Report
Please find my comments attached in the pdf.

Author Response
We appreciate Your comments, and have inculded them in the revised text. You can see my comments in the PDF attached

Round 2
Reviewer 1 Report
The author's have addressed the issues that I raised
Author Response
With Your comment "The author's have addressed the issues that I raised" I have nothing more to add. The language will be improved in the new revision.